# Advances in the Treatment of Giant Cell Arteritis

**DOI:** 10.3390/jcm11061588

**Published:** 2022-03-13

**Authors:** Santos Castañeda, Diana Prieto-Peña, Esther F. Vicente-Rabaneda, Ana Triguero-Martínez, Emilia Roy-Vallejo, Belén Atienza-Mateo, Ricardo Blanco, Miguel A. González-Gay

**Affiliations:** 1Department of Rheumatology, Hospital Universitario de La Princesa, IIS-Princesa, 28006 Madrid, Spain; efvicenter@gmail.com (E.F.V.-R.); ana6n92@gmail.com (A.T.-M.); 2Chair UAM-ROCHE, EPID-Future, Universidad Autónoma Madrid (UAM), 28006 Madrid, Spain; 3Department of Rheumatology, Research Group on Genetic Epidemiology and Atherosclerosis in Systemic Diseases and in Metabolic Bone Diseases of the Musculoskeletal System, IDIVAL, Hospital Universitario Marqués de Valdecilla, 39008 Santander, Spain; diana.prieto.pena@gmail.com (D.P.-P.); mateoatienzabelen@gmail.com (B.A.-M.); rblancovela@gmail.com (R.B.); 4Department of Internal Medicine, Joint Diseases Research Laboratory, Hospital Universitario de La Princesa, IIS-Princesa, 28006 Madrid, Spain; eroyvallejo@gmail.com; 5Department of Medicine, School of Medicine, Universidad de Cantabria, 39008 Santander, Spain; 6Cardiovascular Pathophysiology and Genomics Research Unit, School of Physiology, Faculty of Health Sciences, University of the Witwatersrand, Johannesburg 2050, South Africa

**Keywords:** giant cell arteritis, temporal arteritis, glucocorticoids, DMARD, methotrexate, tocilizumab, abatacept, ustekinumab, jakinibs, mavrilimumab

## Abstract

Giant cell arteritis (GCA) is the most common vasculitis among elderly people. The clinical spectrum of the disease is heterogeneous, with a classic/cranial phenotype, and another extracranial or large vessel phenotype as the two more characteristic patterns. Permanent visual loss is the main short-term complication. Glucocorticoids (GC) remain the cornerstone of treatment. However, the percentage of relapses with GC alone is high, and the rate of adverse events affects more than 80% of patients, so it is necessary to have alternative therapeutic options, especially in patients with worse prognostic factors or high comorbidity. MTX is the only DMARD that has shown to reduce the cumulative dose of GC, while tocilizumab is the first biologic agent approved due to its ability to decrease the relapse rate and lower the cumulative GC doses. However, apart from the IL-6 pathway, there are other pro-inflammatory cytokines and growth factors involved in the typical intima hyperplasia and vascular remodeling of GCA. Among them, the more promising targets in GCA treatment are the IL12/IL23 axis antagonists, IL17 inhibitors, modulators of T lymphocytes, and inhibitors of either the JAK/STAT pathway, the granulocyte-macrophage colony-stimulating factor, or the endothelin, all of which are updated in this review.

## 1. Introduction

Giant cell arteritis (GCA) is an idiopathic granulomatous vasculitis involving medium and large caliber arteries that causes the inflammation of the vascular wall, with cellular infiltration and thickening of its layers, leading to vascular remodeling and occlusion. GCA is the most common vasculitis among individuals over 50 years old of Northern European ancestry [1,2]. The number of patients diagnosed with GCA by the year 2050 in Europe, North America, and Oceania, is expected to be more than 3 million people, of whom approximately 500,000 will be visually impaired [3].

The typical GCA pattern is characterized by the presence of cranial ischemic manifestations such as headache, scalp tenderness, jaw claudication, and visual symptoms [4,5]. Nonetheless, there are other clinical phenotypes that have been identified in recent years. In a series of 693 patients with GCA, four different phenotypes were recognized: (i) classic or cranial pattern (80% of cases), which carries a high risk of visual ischemic complications; (ii) extra-cranial phenotype (9%), that affects the large vessels (aorta and its major branches), which is manifested by limb claudication or subclavian steal syndrome, carrying an increased risk of long-term vascular complications such as aortic aneurysm, vascular stenosis, and the dissection of the aorta; (iii) fever of unknown origin (9%), characterized by persistent fever, constitutional symptoms and elevated inflammatory markers; (iv) a picture mainly characterized by symptoms of polymyalgia rheumatica (PMR), in which asymptomatic vasculitis is accidentally found by a temporal artery biopsy (TAB), or by imaging examinations [6,7]. Interestingly, these phenotypes are not mutually exclusive, and overlapping patterns can often exist [6]. Furthermore, extracranial large vessel vasculitis (LVV)-GCA often presents with non-specific manifestations, such as refractory PMR and constitutional symptoms [8].

The diagnosis of GCA is usually based on the 1990 American College of Rheumatology (ACR) classification criteria for GCA, which included clinical, laboratory and histopathological data found in TAB [9]. These criteria are useful for the diagnosis of classic clinical patterns, but are less sensitive for the identification of the remaining phenotypes. In recent years, certain imaging techniques, especially ultrasound (US), magnetic resonance imaging (MRI), and positron emission tomography (PET)/computed tomography (CT) with 18F-fluorodeoxyglucose (18F-FDG), have been postulated as very sensitive and useful methods for the diagnosis of GCA (see “Imaging tests in the early diagnosis of GCA” in another section of this issue). In this regard, a group of experts from the European League Against Rheumatism (EULAR) has recently provided a set of evidence-based recommendations for the use of imaging in LVV [10].

Regarding the therapy of GCA, glucocorticoids (GCs) remain the keystone of treatment. However, the percentage of relapses with corticosteroids alone is very high, even higher than 50% during the first year of evolution, especially in those patients with a predominance of PMR symptoms and in the LVV-GCA phenotype, particularly when the GC dose is less than 10–15 mg/day [11]. Therefore, in many patients, the combined use of slow-acting immunomodulatory drugs, such as methotrexate (MTX) or some biologic agents, is often recommended.

Tocilizumab (TCZ), an IL-6 receptor antagonist, is currently the only biologic agent approved for the treatment of GCA. The use of TCZ allows one to reduce the frequency of relapses, as well as the cumulative dose of GCs in patients with GCA [12,13]. However, the appearance of adverse events (AEs) or the failure of this biological agent in some cases has led to the search for other biological agents that may also be effective.

Here, we have carried out a comprehensive review of the most relevant works published in recent years on traditional immunosuppressive (IS) drugs and biologic agents for the treatment of GCA. However, for a better understanding of the new therapeutic targets, it is positive to know the main molecular mechanisms, cell populations, and pathogenic pathways on which the pathophysiology of GCA is based.

## 2. Pathophysiology of Giant Cell Arteritis

The etiopathogenesis of GCA remains incompletely understood, and it is probably related to the interaction of genetic and environmental/infectious factors [14,15,16,17]. In fact, a strong association of GCA with HLA class I and II molecules, particularly with HLA-DRB1*04 alleles, has been described [18,19]. The current hypothesis on the pathogenesis of GCA invokes collaboration between the innate and adaptive immune systems and different compartments of the vessel wall, including endothelial cells (ECs), and vascular smooth muscle cells (VSMCs) [14,17,20] (Figure 1). Histologically, GCA is characterized by a pathological granulomatous inflammatory infiltrate within the vessel wall [21,22].

The earliest changes occur at the adventitia of the wall of large and medium-sized arteries. After activation by unknown danger signals, possibly infectious agents, dendritic cells (DCs) resident in the arterial adventitia mature, produce chemokines such as CCL19 and CCL21, and express the co-stimulatory molecules CD83/CD86 required for their interaction with CD4+ T cells coming from the vasa vasorum of the vessel wall. In addition, CCL19 and CCL21 magnify the inflammatory reaction at the adventitia by recruiting and retaining more and more dendritic cells. DCs also release other cytokines, such as interleukin (IL)-1β, IL-6, IL-23 and IL-21 or IL-12 and IL-18, which respectively divide into two distinct immune cell networks [14,20,23,24] (Figure 1). The first one drives the differentiation of activated T cells into Th17 cells; the second one drives Th1 cell formation. Th-17 lymphocytes produce mainly IL-17, IL-21, IL-22, IL-23, and colony-stimulating factor (CSF)-2, which subsequently result in the recruitment of innate immune cells (monocytes, fibroblasts and natural killers), the proliferation of local mesenchymal cells, and the secretion of acute phase reactants (APR) by hepatocytes [20,23]. Th-1 lymphocytes produce mainly interferon-γ (IFN-γ) and tumor necrosis factor α (TNF-α) that favor the production of adhesion molecules, other chemokines (CCL2, CXCL9, CXCL10 and CXCL11) and growth factors (GF), such as platelet-derived growth factors (PDGF), vascular endothelial growth factor (VEGF), and fibroblast growth factor-2 (FGF2). Activated macrophages release pro-inflammatory cytokines such as TNF-α, IL-1 β, and IL-6, thus amplifying the inflammatory response. Finally, macrophage fusion results in the formation of multinucleated giant cells, which contribute to the development of granulomas at the intima-media junction (Figure 1).

Giant cells and activated macrophages release reactive oxygen species (ROS), reactive nitrogen intermediates, and matrix metalloproteinases (MMP)-2 and MMP-9, to promote tissue injury and intimal hyperplasia [25,26]. Activated macrophages and injured VSMCs release PDGF, VEGF, and FGF2, which promote vascular remodeling, neoangiogenesis, and the deposition of extracellular matrix proteins, finally resulting in the luminal stenosis of medium-sized and large arteries [27].

It is currently believed that the Th-17 pathway is mainly involved in the acute inflammatory phase, whereas Th-1 contributes to the inflammatory changes observed in the chronic phase, causing late vascular complications [16,28,29]. It is interesting to remark that, while the Th-17 pathway responds favorably to GC, the Th-1 network is not satisfactorily inhibited by these drugs, with ongoing intima hyperplasia and vascular remodeling [30]. This suggests that GC alone are not sufficient to abolish GCA in the long term, and again reiterates the need for new therapeutic options. Indeed, pro-inflammatory cytokines such as IL-1β, IL-6, IL-12, IL-17A, IL-23, IFN-γ and TNFα observed in GCA patients represent potential targets for the treatment of this intriguing disease [2,31] (Figure 1), and are overproduced.

The importance of the Janus kinase (JAK)/signal activation transducer (STAT) pathway in the pathophysiology of GCA has drawn attention more recently. It is postulated that JAK-1 and JAK-2 inhibition can downregulate both the Th-17 and Th-1 pathways, suppressing the effects of IFN-γ, IL-12, and IL-23 (the targets of ustekinumab (UST)), and IL-6 (the target of TCZ) [32]. As a result, JAK inhibitors are currently under investigation for future GCA therapy.

Furthermore, Samson et al. demonstrated that CD8+ T-cell infiltrate in temporal arteries from GCA patients has a prognostic value suggesting that CD8+ T-cells are recruited within the vascular wall through an interaction between CXCR3 and its ligands, displaying a restricted TCR repertoire in GCA patients [33].

Finally, we want to underline the role of regulatory T cells (Tregs) in the pathogenesis of GCA. Some failure in the quantity and quality of Tregs has been involved in the pathogenesis of several autoimmune diseases. Emerging data suggest that Treg deficiencies are disease specific, affecting distinct pathways in the different vasculitides [34]. In this line, mechanistic studies have identified defective CD8+ Tregs in GCA. More specifically, aberrant signaling through the NOTCH4 receptor expressed on CD8+ Treg cells leads to the redirecting of intracellular vesicle trafficking and failure in the release of immunosuppressive exosomes, ultimately boosting inflammatory attack to medium and large arteries [34].

## 3. Treatment of Giant Cell Arteritis

Since GCA is a treatable disease, the ultimate goal of GCA treatment is to achieve remission. This implies a complete control of the disease without relapses or long-term complications, with a normal quality of life for the subject who suffers it, and ideally with the lowest iatrogenic [35].

### 3.1. Drugs Currently Used in the Management of GCA

#### 3.1.1. Glucocorticoids

Glucocorticoids (GCs) constitute a keystone in the management of GCA for both the induction and maintenance of remission. GCs are very useful for inhibiting the Th-17 response, reducing anemia, systemic symptoms, acute inflammatory response, and the elevation of APR. However, they are not as effective in suppressing the Th-1 cell network, which is responsible for long-term vascular complications [16,28,29].

Current treatment regimens consist of a high initial dose of GCs followed by a gradual taper, with the therapeutic goal of achieving and maintaining clinical remission. Most clinical guidelines, including the EULAR 2018 recommendations for the treatment of LVV, support, starting with a daily dose of prednisolone/prednisone of 40–60 mg, which will be reduced over the next 2–3 months up to a dose of 15–20 mg daily, with the ultimate goal of reaching ≤ 5 mg prednisolone daily after one year [36]. The 2020 British Society of Rheumatology (BSR) guidelines also recommend an initial dose of 40–60 mg of prednisolone daily, tapering to complete GC cessation over a period of 12–18 months [37]. When there are ocular symptoms or visual loss threats, pulses of 500–1000 mg of intravenous (i.v.) methylprednisolone are recommended for 3–5 consecutive days prior to oral prednisolone administration.

The pattern of the GC taper is not well established, and relapses are common during tapering and after the discontinuation of therapy, with a recent meta-analysis demonstrating an overall prevalence of relapses of around 47% in those treated with GC monotherapy [38]. Nowadays, there are no clear predictors of relapse, with the presence of an inflammation in the aorta, and great vessels detected by angio-CT or PET-scan being the only ones currently recognized [39]. Relapses occur most frequently during the first year, when the prednisone dose falls below 10 mg/day, and are rare with prednisone doses above 20 mg/day. Relapses are generally managed by increasing the dose to the previous level that controlled symptoms. Importantly, in about one third of patients, it is not possible to reduce the GC dose to acceptable levels (“GC-dependent patients”).

Glucocorticoid therapy is associated with significant toxicity in over 80% of patients [40,41,42]. Age > 75 years, treatment duration > 2 years, past medical histories of diabetes were risk factors associated with GC-related side effects [42]. The high risk of GC-related AEs in GCA has made it necessary to search for new therapeutic targets.

#### 3.1.2. Methotrexate

Methotrexate (MTX) is the most common conventional IS drug used as a GC-sparing agent. Three randomized controlled trials of MTX as an adjunctive therapy to GCs have been reported to date [43,44,45]. In two of them, no significant differences were found with the addition of MTX to GCs, while in the third one, MTX demonstrated a reduced relapse rate, as well as a long-term cumulative GC-dose. It is important to remark that all three trials were underpowered due to small sample sizes and the relatively low doses of MTX used [43,44,45].

A meta-analysis of these three-randomized placebo-controlled trials yielded a modest role of MTX (10–15 mg/week) to reduce the frequency of relapses and the total prednisone dose [46]. In fact, this analysis reported lower relapse rates (hazard ratio (HR) 0.65, *p* = 0.04), lower cumulative GC doses (mean -842mg at 48 weeks), and a higher rate of GC-free remission (HR 2.8, *p* = 0.001) with MTX [46].

Although MTX is generally a well-tolerated and safe medication, its potential toxicity in the elderly should always be taken into account, especially if there is an associated decrease in renal function. Based on the current available data, the BSR and EULAR guidelines give a conditional recommendation for the use of MTX in GCA. They recommend that MTX might be considered for GCA, in combination with a GC taper, in patients at high risk of GC toxicity, those who relapse, or for patients unable to use TCZ due to recurrent infections, a history of gastrointestinal perforations or diverticulitis, and high cost [36,37]. The 2021 ACR Guidelines for the Management of GCA recommend the use of GCs alone, MTX with GCs or TCZ + GCs as an initial treatment for newly diagnosed GCA, based on the physician’s experience and the patient’s clinical condition, comorbidity, values, and preferences [47]. Although studies with higher doses of MTX used earlier in diagnosis would be necessary, it is likely that the use of MTX will decline in favor of TCZ in the coming years, when TCZ biosimilars become available.

#### 3.1.3. Other Conventional Synthetic Disease-Modifying Anti-Rheumatic Drugs (csDMARD)

With regard to the use of other conventional immunomodulatory agents, a double-blind (DB), randomized, placebo-controlled study in 31 patients with GCA or PMR showed that azathioprine use led to lower GC requirement over the course of one year of therapy [48].

Although a study including 12 patients with PMR and 11 with GCA pointed out a potential efficacy of leflunomide as a GC-sparing agent [49,50], experience with this drug in GCA is scarce. Cyclosporine A, hydroxychloroquine, or dapsone did not show any beneficial effects as GC-sparing agents in patients with GCA [51,52,53,54]. In this regard, a meta-analysis assessing the efficacy of different csDMARD showed that prednisone/prednisolone alone is, in most cases, not inferior in terms of the efficacy and safety with GCs with adjunctive csDMARDs in patients with GCA [51]. A French study of 103 patients with GC-dependent or GC-resistant GCA demonstrated efficacy with cyclophosphamide treatment, since >50% reached long-term remission with a significant GC reduction and a regression of vascular activity on 18F-FDG-PET [55]. However, the AE profile of cyclophosphamide in this setting advises against its use as a GC-sparing therapy. In a retrospective cohort study conducted in 37 patients (65% female) with LVV-GCA prescribing mycophenolate derivatives (mycophenolate mofetil or mycophenolic acid) at diagnosis and followed up for ≥2 years, relapse rates at 1 and 2 years were 16.2 and 27%, respectively, and CRP levels at 1 and 2 years were both 4 (interquartile range: 4–4) mg/L [56].

Taking all this into consideration, the BSR and EULAR guidelines state that there is no sufficient evidence at present to recommend any oral IS drug other than MTX in GCA therapy [36,37].

#### 3.1.4. IL-6 Blockade: Tocilizumab

Tocilizumab (TCZ) is a humanized monoclonal antibody directed against the IL-6 receptor. The hypothesis that IL-6 could play a relevant role in the pathogenesis of GCA encouraged the use of TCZ to treat these patients [57]. Case reports and observational studies of a small series of cases suggested that TCZ could be effective in both newly diagnosed and relapsing patients with GCA by finding a rapid clinical response within 1–2 months of treatment, the normalization of APR, and a decrease in the cumulative dose of prednisone [58,59,60,61]. However, in two multicenter retrospective open-label studies including 22 and 34 refractory or severe GCA and/or with unacceptable GC-related side effects, respectively [60,61], TCZ had to be withdrawn in 3 patients each, due to serious AEs (SAEs) and one death occurred in each study, all possibly drug-related (Table 1) [12,13,39,60,61,62,63,64,65,66].

The efficacy and safety of TCZ for the treatment of both newly diagnosed and refractory/relapsing GCA have been demonstrated in two randomized, DB, placebo-controlled trials [12,13]. In fact, TCZ reduced the total number of relapses and the cumulative dose of GCs compared to GCs administered alone, without increasing SAEs (Table 1).

The first randomized clinical trial was a single-center, phase 2, double-blind, placebo-controlled study that included 30 patients with GCA (23 of new diagnosis and 7 with relapsing disease) [12]. The patients were randomized to receive i.v. TCZ at a dose of 8 mg/kg every 4 weeks plus prednisolone (*n* = 20 patients) or placebo infusion every 4 weeks, plus prednisolone in the rest of the patients (*n* = 10). The primary endpoint was the percentage of patients who reached complete remission at a prednisolone dose of 0.1 mg/kg/day at week 12. Notably, 85% of the 20 GCA patients treated with TCZ experienced a complete remission, versus only 40% of the placebo patients at week 12 (*p* = 0.03) [12]. Moreover, relapse-free survival at 52 weeks was significantly higher in the TCZ-treated group than in the placebo group (85% vs. 20%; risk difference 65%; *p* = 0.001). GCs were rapidly tapered and discontinued by 36 weeks after the TCZ onset. The cumulative prednisolone dose was significantly reduced in the TCZ-treated group at 52 weeks (43 mg/kg) compared to placebo (110 mg/kg) (*p* = 0.0005). Moreover, patients from the placebo group suffered more SAEs than those treated with TCZ (50% vs. 35%) [12]. This study demonstrated that TCZ was effective in inducing remission, preventing relapse, and achieving a reduction in the cumulative dose of GCs. However, CRP and clinical response were considered together as a combined endpoint, which could have overestimated the actual number of remissions, due to the favorable effect of TCZ on CRP [12] (Table 1).

The beneficial effect of TCZ in GCA was supported by data from the GiACTA trial [13]. This phase 3 double-blind, placebo-controlled trial confirmed a TCZ-mediated GC-sparing effect. This study included 251 patients (119 newly diagnosed and 132 relapsing patients) from 14 countries for 22 months. Patients were divided into 4 arms: (1) a weekly dose of subcutaneous (s.c.) TCZ (162 mg), plus a 26-week gradual reduction in prednisone dose; (2) s.c. TCZ (162 mg) administered every other week (eow) along with a 26-week taper of prednisone; (3) weekly s.c. placebo along with a 26-week taper of prednisone; and (4) weekly s.c. placebo, plus a 52-week taper of prednisone. At week 52, patients treated with weekly or eow TCZ achieved sustained remission without GCs in 56% and 53%, respectively, while sustained GC-free remission percentages in the placebo plus 26-week or 52-week taper of prednisone were 14% and 18%, respectively (*p* < 0.001 in both cases) [13]. In addition, a sensitivity analysis showed that, after excluding CRP, sustained remission rates were 59% for the TCZ weekly group and 55% for the TCZ eow arm, respectively, much higher than in the prednisone plus placebo groups.

Relapses were less frequent in patients treated with TCZ (23% and 26% in those who received TCZ every week and eow, respectively) than in those included in the placebo arms of 26 and 52 weeks of prednisone taper (68% and 49%, respectively). In addition, TCZ-treated patients achieved a significant GC-sparing effect. Importantly, TCZ-treated patients had fewer SAEs than placebo-treated patients [13]. Based on these results, both the United States Food and Drug Administration (FDA) and the European Medicines Agency (EMA) approved the weekly use of s.c. TCZ in GCA, making TCZ the only biologic agent currently approved for the treatment of this disease. In line with data from GCA, open-label studies have also shown that TCZ is effective in patients with isolated aortitis, and in Takayasu´s arteritis [67,68].

Recent real-life observational studies are of great interest because they include patients who are routinely excluded from clinical trials. In a Spanish multicenter retrospective study of 134 refractory GCA patients treated with either monthly administered i.v. TCZ 8 mg/kg (*n* = 106) or weekly s.c. TCZ 162 mg (*n* = 28) plus GC, sustained clinical remission was achieved in 55.5%, 70.4%, 69.2%, and 90% of patients at 6, 12, 18, 24 months, respectively [64] (Table 1). Most patients (73.1%) had received conventional or biologic (b)DMARDs before starting TCZ treatment, and 38.8% received TCZ in combination with csDMARDs, mainly MTX. Of note is the rapid significant clinical response in 93.4% of patients after a month of treatment with TCZ, regardless of the time of evolution of GCA, the route of administration of TCZ, or the initial dose of prednisone. A comparative study of these patients with patients in the GiACTA trial showed that TCZ was equally effective in this clinical practice population, despite older age, longer disease duration, higher ESR values, and the greater use of csDMARDs [64]. However, the rate of serious infections (11.9%; 10.6/100 patient-years) was higher than in the GiACTA study and clinical trials in rheumatoid arthritis, especially in patients with higher doses of prednisone in the first 3 months of treatment (*p* = 0.003).

Evidence of the ability of TCZ to reverse blindness is scant [60,61]. However, the incidence of vision loss during TCZ treatment is very low. In a retrospective, single-center analysis of 60 patients with GCA treated with i.v. or s.c. TCZ, a reduced incidence of visual manifestations and no new cases of permanent visual loss occurred while patients were receiving TCZ [69]. Consistent with this, in our experience with 471 TCZ-treated GCA patients followed up in a real-world setting, ocular manifestations stabilized after the initiation of TCZ, with no new episodes of blindness detected after the initiation of TCZ (observations not published). Conversely, in another observational single center study of 186 GCA patients treated with TCZ, Amsler et al. described two cases of blindness (1.1%) during TCZ treatment, but both occurred at the start of TCZ treatment [66]. Therefore, in patients starting TCZ, a rapid GC decrease is not recommended, at least initially, especially when visual symptoms are present.

Once the efficacy and safety of TCZ for the treatment of GCA have been demonstrated, many unanswered questions remain: Could TCZ be as effective as monotherapy as in combination with GC? What is the best starting dose and GC taper schedule? Does the association of csDMARDs such as MTX with treatment with TCZ provide added value? How long should TCZ be administered? Can we optimize TCZ, and at what speed? Do AEs increase with duration of treatment? Are there biomarkers that help predict relapse after treatment with TCZ? Is TCZ effective in reducing visual disturbances? What is the role of biomarkers and imaging in the follow-up of patients treated with TCZ? Real-world and post hoc studies, as well as open-label extension phases of randomized clinical trials, are beginning to provide clues to answering these questions.

Most studies to date have used TCZ in combination with prednisone/prednisolone, so it is difficult to obtain robust results on its use in monotherapy, although there are data suggesting that TCZ in monotherapy may be equally useful in newly diagnosed patients with LVV (GCA or Takayasu’s arteritis) when started early from diagnosis [70]. However, as many flares in the GiACTA trial occurred while patients were taking >10 mg/day prednisone [71], and major relapses with visual disturbance have been reported with TCZ between week 11 and 24 when patients were on low-dose prednisone (0.1 to 0.11 mg/kg/day) [12,13], the combination with GC seems reasonable, since it provides greater safety and a faster onset of action, especially if there are visual or major organ complications.

The optimal starting dose of GC to treat GCA is still unknown. Most patients in published studies have received an initial dose of prednisone between 20 mg and 60 mg daily, although i.v. methylprednisolone boluses have also been used in some patients in clinical practice. While in the study by Calderón-Goercke et al. [64], response to TCZ did not appear to be influenced by the initial dose of GC, a post hoc analysis of the GiACTA trial has shown that patients with lower basal doses of prednisone (≤30 mg/d) had a higher risk of failure to TCZ than those who received >30 mg/d with an odds ratio (OR) of 2.4 (95% CI: 1–5.9; *p* = 0.046) [72].

GC-related AEs and the rapid response found with TCZ have led to the exploration of shorter GC withdrawal regimens. Nannini et al. [63] treated 15 GCA patients with thoracic LVV, both newly diagnosed and recurrent, with i.v. TCZ (*n* = 9) or s.c. TCZ (*n* = 6), plus a GC tapering regimen for 2 months, achieving sustained remission in all patients on TCZ treatment. This short GC regimen could be especially useful in patients with comorbidities such as diabetes, hypertension, osteoporosis, or glaucoma (Table 1).

Regarding the usefulness of adding MTX to TCZ treatment, a preliminary post hoc analysis of the GiACTA trial suggests that add-on MTX may not increase the likelihood of sustained remission, reduce disease relapse rate, or improve GC sparing in patients with GCA, although this should be confirmed in larger studies [73]. Real-world data did not find significant differences in SAEs or serious infections between patients, with or without add-on MTX [64].

Regarding the optimal duration of treatment with TCZ, it remains an unresolved issue. Short TCZ courses have also been used. In the series by Régent et al. [61], TCZ withdrawal occurred after a mean treatment duration of 5.6 months in 23 patients (planned in 20, due to AEs in 3), and 8 of these patients relapsed after a mean of 3.5 months. Five of them showed clinical and biological responses to TCZ reintroduction. A prospective multicenter open-label study included 20 new-onset or relapsing GCA patients that received four monthly i.v. infusions of TCZ 8 mg/kg from weeks 0 to 12, plus GC tapered over 52 weeks [39]. Relapse-free survival at week 52 was 45%, and most patients relapsed after a median of 28 weeks.

TCZ (i.v. or s.c.) tapering from month 6 to month 10, followed by the discontinuation of TCZ at month 12, has also been investigated in a prospective open-label study [63]. At month 18, 66.7% of patients remained in remission after TCZ withdrawal. Five patients relapsed 2–4 months after stopping TCZ, but recovered remission after reintroducing TCZ at the last dose administered, without GC. In the GiACTA trial, many flares occurred while patients were taking >10 mg/day prednisone (25% and 22% in the TCZ and placebo groups, respectively), and APR levels were not reliable indicators of flares in either arms [71].

Long-term open-label extensions of both TCZ randomized clinical trials highlight that sustained treatment-free remission is possible in patients that received TCZ for 52 weeks. In the open label extension of the trial by Villiger et al. [12], half of the 17 patients of the TCZ arm that were in remission at 52 weeks maintained DMARD-free remission after a mean follow-up of 28.1 months [62]. Most of the patients relapsed within the first 5 months after TCZ withdrawal, and two of them after 13 and 14 months, respectively. In the 2-year extension study of the GiACTA trial, among the 81 patients of the weekly TCZ arm that were in complete remission at week 52, nearly half of them (47%) maintained remission for the entire extension period [74]. A higher proportion of treatment-free patients and a lower cumulative dose of GC were observed in patients originally assigned to TCZ than to PBO. Retreatment with TCZ (with or without GC) restored complete remission.

In a recent meta-analysis by Berti et al. on 10 randomized clinical trials (9 phase 2 and 1 phase 3), TCZ, i.v. GCs, and MTX significantly improved the likelihood of being relapse-free with relative risks (RR) (95%CI) of 3.54 (2.28–5.51), 5.11 (1.39–18.81), and 1.54 (1.02–2.30), respectively [75].

However, despite the benefits of TCZ treatment, relapse is still frequent following TCZ discontinuation. Several potential predictors of recurrence after the discontinuation of TCZ have been identified: GCA duration ≥ 3 months at the start of TCZ therapy [71]; the presence of baseline aortitis, CRP > 70 mg/L or Hb ≤ 10 g/dL [39]; younger age and more locations of mural enhancement in magnetic resonance angiography (MRA) at TCZ initiation [62]; and cranial symptoms (compared to PMR symptoms) in patients with new-onset disease at the time of TCZ start [76].

Although the majority of flares that occurred during TCZ therapy or after discontinuation were not severe, visual manifestations were rare at the time of the flare, and no cases of blindness were described after TCZ withdrawal; tight control must be ensured, especially because ESR was normal in one-third of the patients who experienced flares [76].

Long-term use may be necessary for the continued maintenance of remission. Consequently, in clinical practice, an extension of TCZ treatment to 18–24 months could be recommended. If clinical remission is maintained and GCs are successfully tapered, TCZ dose reduction (e.g., lower subcutaneous dose) may be considered for the next 6 to 12 months before complete discontinuation [77]. Interestingly, most AEs with TCZ occur in the first six months of treatment. In fact, in patients treated for >1 year, there was no increased incidence of AEs compared with patients treated for <12 months [78]. Similar results were obtained by Calderón-Goercke et al., which found no significant differences in SAE, and serious infections between patients who have received TCZ for >12 months, compared to those treated for ≤12 months [64]. Currently, there is an extension study (ClinicalTrials.gov Identifier: NCT03202368) investigating the long-term safety of s.c. TCZ in patients with GCA who have flares or persisting disease activity.

Regarding prognostic factors of response to TCZ, a post hoc analysis of the GiACTA study, after adjustment for confounders, showed that the strongest risk factors of therapeutic failure in GCA were treatment with prednisone alone and female sex, while lower prednisone doses at the beginning of therapy and impaired patient-reported outcomes were associated with treatment failure in the TCZ-arm [72].

There are still no definite data supporting the usefulness of new biomarkers in the follow-up of GCA patients treated with TCZ. Nevertheless, regarding the classic APR (ESR and CRP), it should be noted that their usefulness may be masked in the follow-up of these patients, given the positive impact that TCZ has on the elevation of these two biomarkers, in such a way that there may be relapses of the disease with normal APR [71], which further reinforces the value of the symptoms and the physician’s opinion on the evolution of these patients.

The role of imaging to monitor treatment response and predict relapse is still pending elucidation. A significant correlation between some PET/CT parameters and the absence of response to TCZ has been suggested by some authors [63], as well as the association of more active mural enhancement areas in MR-angiography (MRA) with relapse [62]. However, although a tendency to a reduction of enhancing areas in MRA was observed in patients in remission, all the patients had MRA mural signal at the end of the follow-up, despite being in remission [62]. These results might reflect the need of longer periods of TCZ treatment in patients with LVV.

In the only multicenter, randomized, double-blind, placebo-controlled, parallel-group study, two-part phase 3 trial (NCT02531633; Part A [52-week DB treatment]; Part B [104-week follow-up]) to evaluate the efficacy and safety of sirukumab, a selective, high-affinity human IL-6 monoclonal antibody, in the treatment of GCA, the proportion of patients with flares (week 2–52) was lower with sirukumab (18.4–30.8%) than placebo (37.0–40.0%), with no unexpected safety findings [79]. However, this study was terminated early (October 2017) by sponsor decision. Currently, we have no further information on the efficacy of this drug.

The same occurred with sarilumab, whose pivotal study (NCT03600805) was interrupted early due to the COVID-19 pandemic.

### 3.2. Role of Non-Tocilizumab Biologic Disease-Modifying Anti-Rheumatic Drugs

#### 3.2.1. TNF-α Antagonists

TNF-α inhibitors have been used in patients with rheumatic diseases refractory to conventional therapies. They were used in patients with GCA. However, the efficacy was poor in most cases. In this regard, Hoffman et al. conducted a phase II, DB, randomized, placebo-controlled study to determine the efficacy of infliximab (IFX) (a chimeric monoclonal antibody) in 44 patients with new-onset GCA [80]. Patients were randomized to receive placebo (*n* = 16) or IFX at a dose of 5 mg/kg/infusion (*n* = 28), at baseline and at weeks 2, 6, 14, 22, 30, 38, and 46. No differences between IFX-treated and placebo-treated patients in terms of relapse-free patients (43% IFX vs. 50% placebo) or cumulative doses of prednisone were observed at 22 weeks. Furthermore, the frequency of infections was higher in the group treated with IFX [80] (Table 2).

Another trial was performed with the anti-TNF-adalimumab (a fully human monoclonal antibody). Patients with newly diagnosed GCA were randomized to receive adalimumab (*n* = 34) or placebo (*n* = 36). However, adalimumab did not show superiority in terms of remission with less than 0.1 mg/kg prednisone at 6 months. Serious adverse events occurred in five (14.7%) patients on adalimumab and 17 (47.2%) on placebo, including serious infections in 3 patients on adalimumab, and 5 on placebo [81] (Table 2).

The only positive results were found in a multicenter, randomized, DB, placebo-controlled trial that included 17 patients treated with the soluble TNF receptor fusion protein etanercept [82]. Although the results were not significant, the clinical control of the disease without GC at 52 weeks was 50% in the ETN group compared to 22.2% in the placebo group (Table 2) [82].

Taken together, and based on current evidence, anti-TNF therapy is generally not recommended in patients with GCA.

#### 3.2.2. IL-12/IL-23 Pathway Inhibition

IL-12 and IL-23 are two cytokines central to the inflammatory and proliferative pathways of GCA (Figure 1) [83,84]. Ustekinumab (UST) is a therapeutic human immunoglobulin (Ig) G1 kappa clonal antibody that binds to the p40 subunit common to both unbound IL-12 and IL-23. Therefore, the blockade of these two cytokines prevents IL-12 and IL-23 mediated effects, including Th1 and Th17 responses, which have been recognized as key players in the pathogenesis of GCA [83,84,85] (Figure 1).

So far, separate groups have conducted two open-label single-center prospective trials, showing conflicting results [29,86,87]. The first one, by Conway et al., evaluated the efficacy and safety of UST 90 mg, subcutaneously administered at weeks 0, 4, and then every 3 months, in 14 patients with refractory GCA, and a median follow-up period of 13.5 months [29] (Table 2). The primary outcome was the comparison of the median GC dose required to control the disease prior to UST, and at last follow-up. No patient had a relapse of GCA while receiving UST. A significant reduction of GC dose was observed (*p* = 0.001); 29% discontinued GC and 92% discontinued other csDMARDs. Seven patients had LVV on CT angiogram prior to UST. Repeat imaging performed in five of these seven, after a median duration of 13 months, showed no new stenosis or aneurysms. Three patients stopped UST due to AEs, two of whom subsequently had flares of PMR [29] (Table 2). Similar results were subsequently obtained on a sample of 25 patients by the same authors [86]. UST 90 mg was subcutaneously administered every 12 weeks. The follow-up in this case was 52 weeks. Six patients (24%) discontinued GC, while 76% discontinued other csDMARDs. CT angiography demonstrated an improvement of LVV in all patients studied. No unexpected AEs were observed [86].

However, a prospective, single-center, open-label trial of UST in 13 patients with recent-onset or recurrent active GCA by Matza et al. did not give favorable results [87]. Patients received a 24-week prednisone taper and s.c. UST 90 mg at baseline and at weeks 4, 12, 20, 28, 36, and 44. Only three patients (23%) achieved the primary endpoint. Of the 10 patients (77%) who failed to achieve the primary endpoint, seven relapsed after a mean period of 23 weeks [87]. Nonetheless, the design of this study has drawn much criticism [88,89]. In fact, there was a higher relapse rate than in clinical practice. This could be due to a rapid reduction in GC. Furthermore, an effective treatment such as UST with a slow onset of action may falsely appear ineffective with this design. Therefore, the rapid GC reduction in this study may have biased the results to null [89].

**Table 2 jcm-11-01588-t002:** Role of non-tocilizumab biologic disease-modifying antirheumatic drugs in giant cell arteritis.

Author, Year [Ref.]	Type of Study	*n*	Sex F (%)	Population/Median Follow-Up	Therapeutic Protocol	Prior DMARD n (%)	Main Efficacy Outcomes	Serious Adverse Events
TNF-α Antagonists								
Hoffman et al., 2007 [80]	Multicenter, randomized, DB, PBO-controlled trial	44 (IFX 28, PBO 16) 2:1 ratio	IFX 24 (86) PBO 11 (69)	Newly diagnosed GCA in GC-induced remission. FU: 54 weeks (early termination after interim analysis at week 22)	i.v. IFX 5mg/kg vs. PBO	Prior DMARD not allowed	Differences in relapse-free patients (43% IFX vs. 50% PBO) and % of patients with GC tapering without relapse (61% IFX vs. 75% PBO) at 22w between groups were NS	29% IFX vs. 25% PBO, NS Serious infections 11% IFX vs. 6% PBO, NS
Martinez-Taboada et al., 2008 [82]	Multicenter, randomized, DB, PBO-controlled trial	17 (ETN 8, PBO 9) 1:1 ratio	ETN 6 (75) PBO 8 (88.9)	Clinically asymptomatic biopsy-proven GCA with GC-related comorbidity. FU: 52 weeks	s.c. ETN 25 mg twice weekly vs. PBO	Prior DMARD not allowed	Clinical disease control without GC at 52 w: 50% ETN vs. 22.2% PBO (NS). ETN cumulative GC dose was significantly lower (*p* = 0.03)	12.5% ETN vs. 11.1% PBO, NS
Seror et al., 2018 [81]	Multicenter, randomized, DB, PBO-controlled trial (HECTHOR trial)	70 (ADA 34, PBO 36)	ADA 24 (71) PBO 28 (78)	Newly diagnosed GCA (GCA-related visual symptoms were an exclusion criteria). FU: 10 weeks	s.c. ADA 45 mg q2w vs. PBO	Prior DMARD not allowed	Remission on less than 0.1 mg/kg of prednisone at week 26: 59% ADA vs. 50% PBO (NS). NS differences in prednisone dose reduction or % of relapse-free patients	14.5% ADA vs. 47.2% PBO. Serious infections: 3 ADA vs. 5 PBO. Deaths: 1 ADA (pneumonia) vs. 2 PBO (septic shock and cancer)
Ustekinumab								
Conway et al., 2016 [29]	Single center, prospective open-label registry	14	11 (79)	Refractory GCA (inability to taper GC to <10 mg/d due to symptoms of active GCA with a minimum of two relapses). FU: 13.5 months	s.c. UST 90 mg every 3 months	12 (86)	No relapses. Significant reduction of GC dose (*p* = 0.001). 29% stopped GC and 92% stopped other DMARD. Image improvement in LVV (*n* = 5), without new stenoses or aneurysm	3 Discontinuations due to AE
Conway et al., 2018 [86]	Multicenter, open-label prospective registry	25	20 (80)	Refractory GCA (inability to taper GC due to recurrence of symptoms consistent with active GCA, after an initial treatment response to high-dose GC). FU: 52 weeks	s.c. UST 90 mg every 3 months	17 (68)	No relapses. Significant reduction of GC dose (*p* < 0.001) and CRP decrease (*p* = 0.006). 24% stopped GC and 76% stopped other DMARD. Image improvement in LVV (*n* = 8), without new stenoses or aneurysm	3 Discontinuations due to AE: 1 recurrent respiratory tract infections, 1 alopecia and 1 non-dermatomal limb paresthesia
Matza et al., 2021 [87]	Single center, single-arm prospective open-label trial	13	11 (85)	Active new-onset (39%) or relapsing GCA. FU: 52 weeks (enrollment closed prematurely due to relapse of 7/10 initial patients)	s.c. UST 90 mg every 3 months	2 (15)	23% achieved prednisone-free remission (absence of relapse through week 52 and normalization of ESR and CRP). 7 Patients relapsed after a mean period of 23 w	1 SAE: mild diverticulitis that required hospitalization
Abatacept								
Langford et al., 2017 [90]	Multicenter, randomized DB, PBO-controlled study	41 (20 ABA, 21 PBO) 1:1 ratio	ABA 16 (80) PBO 21 (100)	Newly-diagnosed or relapsing GCA with active disease within the prior 2 m. FU: 12 months	Initially (*n* = 49): i.v. ABA 10 mg/kg/m At 12 w (*n* = 41): DB randomization to ABA vs. PBO of cases in remission	NR Prior bDMARD was not allowed within established time schedule	Relapse-free survival at 12 m: 48% ABA vs. 31% PBO (*p* = 0.049). Longer duration of remission with ABA (9.9 m) vs. PBO (3.9 m), *p* = 0.023	23 SAE in 15 patients. NS difference in frequency or severity of AE between treatment arms, including the rate of infection or SAE. No deaths
Rossi et al., 2021 [91]	Single center, two-arm prospective open-label study	33 (17 TCZ, 16 ABA) 1:1 ratio	21 (63.6)	Consecutive biopsy-proven newly diagnosed or relapsing GCA. FU: 12 months	TCZ: i.v. 8mg/kg/m (*n* = 8), s.c. 162 mg/w (*n* = 9) ABA: s.c. 125 mg/w Combination with other DMARD was not allowed	22 (66.6)	i.v. TCZ, s.c. TCZ and ABA clinical response was complete in 57%, 67% and 31%, and partial in 43%, 16% and 31%, respectively 100% TCZ group and 43% ABA group reduced prednisone to ≤ 7.5 mg/d at 12 m, *p* = 0.0003. Switch due to inefficacy more frequent with ABA (0.0445)	No significant side effects
Sirukumab								
Schmidt et al., 2020 [79]	Multicenter, randomized DB, PBO-controlled parallel-group study + open-label extension	161 (SIR 107, PBO 54)	124 (77)	Newly diagnosed or relapsing GCA. FU of DB phase: 52 weeks FU of OL phase: 104 weeks (early termination by sponsor decision)	DB phase: s.c. SIR 100 mg q2w + 6 m or 3 m of GC taper; s.c. SIR 50 mg q4w + 6 m GC taper;PBO q2w + 6 m or 12 m GC taper OL phase: SIR 100 mg q2w at investigator discretion	Prior cs- and bDMARD was not allowed within established time schedule	At 52 w: Sustained remission not achieved by 82.4–88.9% patients in SIR arms and 100% in PBO arms; Lower % of flares with SIR than PBO (18.4–30.8% vs. 37–40%); Highest % of flares (23.1%) and withdrawals (61.5%) with SIR 100 mg q2w + 3 m GC taper. OL phase: 60% maintained remission at 4w without SIR administration; No flares	At 52 w: 19.3% SAE; NS differences across arms; No deaths. OL phase: No SAE; No deaths
Meta-analysis								
Berti et al., 2018 [75]	10 RCT (9 phase 2 and 1 phase 3)	645					TCZ, i.v. GC and MTX significantly improved the likelihood of being relapse free with relative risks and 95% CI of 3.54 (2.28, 5.51), 5.11 (1.39, 18.81) and 1.54 (1.02, 2.30)	
Song et al., 2020 [92]	6 RCT (2 TCZ, 3 TNF antagonists 1 and ABA)	260 patients 193 controls					Remission rate higher for TCZ than PBO (OR 7.009, 95% CI 3.854–12.75, *p* < 0.001). Relapse rate lower for TCZ than PBO (OR 0.222, 95% CI 0.129–0.381, *p* < 0.001). NS difference in remission and relapse between groups with TNF antagonists, ABA and PBO	Number of SAE lower for TCZ than PBO (OR 0.539, 95% CI 0.296–0.982, *p* = 0.044). NS difference in SAE among patients treated with TNF antagonists, ABA and PBO. Infection rate higher for TNF antagonists than PBO (OR 2.407, 95% CI 1.168-4.960, *p* = 0.017), but with NS differences between TCZ, ABA and PBO

Abbreviations: ABA: abatacept; ADA: adalimumab; AE(s): adverse event(s); bDMARD: biologic disease modifying antirheumatic drug; CI: confidence interval; CMV: cytomegalovirus; CRP: C-reactive protein; csDMARD: conventional synthetic disease modifying antirheumatic drug; CV: cardiovascular; d: day; DB: double-blind; DMARD: disease modifying antirheumatic drug; ESR: erythrocyte sedimentation rate; ETN: etanercept; F: female; FU: follow-up; GC: glucocorticoids; GCA: giant cell arteritis; GI: gastrointestinal; i.v.: intravenous; kg: kilogram; m: month; LVV: large vessel vasculitis; mg: milligram; MI: myocardial infarction; MTX: methotrexate; n: number; NS: not significant; OL: open-label; PBO: placebo; qw: every week; q2w: every other week; RCT: randomized controlled trial; RD: risk difference; SAE: serious adverse event; s.c.: subcutaneous; SIR; sirukumab; TBC: tuberculosis; TCZ: tocilizumab; TNF: tumor necrosis factor; vs: versus; w: week.

Taking all this in consideration, the results on the efficacy of UST in new or refractory GCA are encouraging, but more studies with appropriate design and larger numbers of patients are needed. Currently, there is a new phase 2, randomized, parallel assignment, open-label study (NCT03711448) with s.c. UST 90 mg, at weeks 0, 4, 12 and 28, plus GC taper versus GC taper in relapsing GCA, with the objective of evaluating patients in remission at week 52 who are still in the recruitment period (Table 3).

Additionally, guselkumab, a monoclonal antibody that targets the p19 sub-unit of IL-23, is currently under evaluation in the phase 2 THEIA trial. This multicenter, randomized, placebo-controlled, DB trial aims to assess the efficacy of guselkumab in combination with a 26-week GC taper versus placebo, with a 26-week GC taper in 60 patients with new onset or relapsing GCA in achieving GC-free remission (Table 3). The predicted completion date is October 2023.

#### 3.2.3. T Cell Modulation: Abatacept

Given the key role of T cell response (Th1, Th17 subsets) in GCA pathogenesis, it is logical to think that the inhibition of this pathway could be a target in the treatment of GCA. Abatacept (ABA), a CTLA4-Ig small molecule fusion protein, binds to CD80/CD86, thus preventing the engagement of CD28 with its ligand, which ultimately inhibits T cell activation (Figure 1).

To date, one randomized control trial (NCT04474847) assessing the efficacy of ABA in the management of GCA has been completed (Table 2) [90]. This is a phase 3, randomized, parallel assignment, DB study comparing s.c. ABA 125 mg every week for 12 months, versus s.c. placebo every week for 12 months in 41 patients with newly diagnosed or relapsing GCA (20 randomized in the ABA arm). In this trial, ABA was superior in increasing relapse-free survival at 12 months versus placebo (48% vs. 31%; *p* = 0.049). The median duration of remission was also higher in the ABA trial compared to placebo (9.9 months vs. 3.9 months; *p* = 0.023) (Table 2). In this study, however, there are some confounders that might potentially favor the risk of relapse in the prednisolone monotherapy group, thus overestimating the efficacy of ABA in comparison [90].

A recent real-world study comparing ABA versus TCZ conducted in 33 patients with biopsy-proven GCA showed a complete or partial clinical response in 62% of patients treated with ABA, versus 88% of those treated with TCZ. However, the percentage of patients receiving prednisone at 12 months, not exceeding 7.5 mg/day as maintenance in any of the groups, was 43% in the ABA group compared to 100% in the group treated with TCZ [91]. These results speak in favor of the possible use of ABA as a rescue therapy in the management of GCA. In contrast, in a recent meta-analysis by Song et al. of six randomized clinical trials, including 260 GCA patients and 193 controls, no significant differences between anti-TNF, ABA, and placebo were found [92] (Table 2).

#### 3.2.4. IL-1 Inhibition

IL-1 is also a potential therapeutic target in the treatment of GCA (Figure 1). This is supported by elevated serum IL-1 levels in patients with GCA, and increased IL-1β mRNA expression in the temporal arteries of subjects with GC-refractory disease [93,94]. Anakinra (ANK), a recombinant form of the human IL-1 receptor antagonist, proved to be effective in the management of GCA in three patients with refractory GCA. In this line, there is a phase 3, randomized, parallel assignment, double-blind study with s.c. ANK 100 mg/d for 16 weeks, versus s.c. placebo every day for 16w (GiAnT; NCT02902731) to analyze the global relapse rate at 26 weeks in new-onset and relapsing GCA, which is still in the recruitment period (Table 3).

In a retrospective study from the French Group for LVV Study, which analyzed GCA patients treated with ANK, all six patients exhibited complete clinical and biological remission after a median duration of ANK therapy of 19 (18–32) months [95]. Among the four patients with large-vessel involvement, one had a disappearance of aortitis under ANK, and three showed a decrease in vascular uptake. After a median follow-up of 56 (48–63) months, GCs were discontinued in four patients, and GC dosage was decreased to 5 mg/day in two. One patient relapsed 13 months after the introduction of ANK in the context of increasing the daily interval of ANK injection to every 48 h [95].

However, a phase II proof of concept study with gevokizumab, another monoclonal antibody inhibitor of IL-1β in GCA, was finished early, due to the sponsor’s decision and low sample size.

### 3.3. Therapeutic Lines under Research

#### 3.3.1. IL-17 Inhibition

IL-17A is a proinflammatory cytokine expressed by cells of the innate immune system, as well as Th17 cells. Secukinumab is a highly selective human IL-17A monoclonal antibody. Currently, there is a case reported of psoriatic arthritis associated with GCA treated successfully with secukinumab [96]. This pathogenic basis has led to the development of two multicenter, randomized, double-blind clinical trials for evaluating the efficacy and safety of secukinumab in GCA. The first one of these studies, a phase 2 trial, completed in June 2021 (TitAIN trial), has only released a study protocol so far [97]. The other one, in phase 3, is still in the recruitment period (Table 3).

#### 3.3.2. Janus Kinase Inhibitors

The Janus kinase/signal transducers and activators of transcription (JAK/STAT) pathway plays an important role in cell regulation in humans. Many different molecules, including interleukins, interferons, and growth factors, among others, use this critical JAK/STAT pathway to transmit their effects through type 1 and type 2 receptors [98,99]. Therefore, the role of the JAK/STAT pathway and, more specifically, its inhibition through JAK small molecule inhibitors (Jakinibs) has attracted much interest in the area of inflammatory and autoimmune disorders, including GCA [99]. In this line, there is growing evidence that the JAK/STAT signaling pathway is involved in the pathogenesis of LVV. Zhang et al. demonstrated an activation of STAT1 and STAT1/2 heterodimer pathway within GCA vascular lesions [100]. In addition, tofacitinib has been shown to effectively inhibit LT proliferation in vessel-wall infiltrates, reduce IFN-Υ and IL-17 production, and reduce neoangiogenesis and intimal hyperplasia in a xenograft experimental model [100].

A case of a 52-year-old woman diagnosed with LVV with overlap features of Takayasu’s arteritis/GCA after the failure of conventional immunosuppressive and biologic therapies -including TCZ and IFX-, who responded successfully to baricitib 4 mg/day, has recently been reported [101]. However, no clinical trials assessing the therapeutic role of Jakinibs in GCA have been published yet. Baricitinib, a jakinib that selectively inhibits JAK1 and JAK2, is currently being evaluated in an open-label phase II pilot study (NCT03026504), examining its role as an adjunct to a standardized GC taper in 15 subjects with relapsing GCA (Table 3). In addition, the SELECT-GCA trial (NCT03725202), a planned phase 3, multicenter, randomized controlled trial examining the role of upadacitinib, a selective JAK1 inhibitor, in combination with a 26-week GC tapering regimen in 420 patients with active GCA, with the aim of evaluating the percentage of patients achieving sustained remission at week 52, is currently under recruitment (Table 3).

#### 3.3.3. Granulocyte-Macrophage Colony-Stimulating Factor

Granulocyte-macrophage colony-stimulating factor (GM-CSF) is a pleiotropic inflammatory mediator implicated in the pathogenesis of GCA [35]. Both GM-CSF and its receptor GM-CSF-α are upregulated in the TAB of patients with GCA [102]. Furthermore, elevated GM-CSF levels have been demonstrated in the peripheral blood of patients with active GCA [103]. Indeed, GM-CSF is produced by Th1 and Th17 lymphocytes, and it exerts its pathogenic potential by different targets in the context of GCA [35] (Figure 1).

GM-CSF stimulates the differentiation of monocytes to DCs, and enables DCs to program naïve CD4+ cells into Th1, Th17, and T follicular helper populations [102,104]. Furthermore, GM-CSF stimulates macrophage differentiation, promotes the activation and formation of giant cells, and catalyzes the proliferation and migration of vascular ECs, all of which are central to the vascular damage caused, remodeling and neoangiogenesis seen in GCA [23].

Mavrilimumab (MAV) is an IgG4 humanized monoclonal antibody that inhibits GM-CSF receptor alpha, acting as a competitive antagonist of GM-CSF activity. Interestingly, MAV decreased intimal thickness, T-cell infiltration, and neo-vessel formation in a human artery-SCID (severe combined immunodeficiency) chimera model [105]. In this line, Corbera-Bellalta et al. have recently demonstrated that MAV reduces infiltrating cells, pro-inflammatory markers and neoangiogenesis in ex vivo cultured arteries from patients with GCA [106]. In fact, MAV reduced the expression of molecules relevant to T-cell activation (HLA-DR) and Th1 differentiation (IFN-γ), the pro-inflammatory cytokines IL-6, TNFα and IL-1β, as well as molecules related to vascular injury (MMP-9, lipid peroxidation products and inducible nitric oxide synthase). Furthermore, MAV reduced CD34+ cells and neoangiogenesis in GCA lesions [106].

Finally, preliminary results from an ongoing phase 2 RCT (NCT03827018) are stimulating (Table 3). Seventy patients with either new onset or relapsing/refractory GCA were randomly assigned to receive either biweekly s.c. MAV 150 mg, or placebo, in addition to a 26-week tapering GC regimen. At week 26, MAV decreased disease flares compared to placebo (19% of patients receiving MAV vs. 46.4% in the placebo group) and improved the rate of sustained remission (83% in the MAV group vs. 49.9% in the placebo) [107]. Results were comparable in both the new onset and relapsing/refractory GCA patient groups. These preliminary results are very promising, and provide a new effective, safe, and tolerable GC-sparing agent in the management of GCA [35].

**Table 3 jcm-11-01588-t003:** Ongoing or yet to publish randomized clinical trials investigating biologic or targeted therapies for giant cell arteritis.

Drug [Therapeutic Regimen ] [Ref.]	Trial Name and Identifier	Target	Duration	Type of Trial and Phase	Control	Population	Target *n*	Primary Outcome	Status (January 2022)
IL-6R antagonists									
Tocilizumab [s.c. TCZ 162 mg/w 52 w + 8 w GC taper]	NCT03726749	IL-6	52 weeks	Phase 4, open-label	None	New-onset and relapsing GCA	30	Sustained remission at w52	Recruiting
Tocilizumab [s.c. TCZ 162 mg/w 4 w + GC taper 18 m (1g iv MP/d 3 d + oral GC) + ASA 75 mg/d vs. GC + ASA 75 mg/d]	TOCIAION NCT04239196	IL-6	18 months	Phase 2, randomized, parallel assignment, open-label, non-comparative	None	AION due to GCA	58	Ocular change at w8	Unknown
Tocilizumab [s.c. TCZ 162 mg/w 52 w + GC versus escalating s.c. MTX up to 0.3 mg/Kg/w 52 w + GC]	METOGiA NCT03892785	IL-6	78 weeks	Phase 3, randomized, parallel assignment, open-label	MTX (≤20 mg/w) + GC	Active GCA within 6 weeks before randomization	200	Percentage of patients alive without relapse after initial remission or deviation from the scheduled regimen of prednisone at w78	Recruiting
Tocilizumab [s.c. TCZ 162 mg/w 24 w versus s.c. PBO/w 24 w]	TOGIAC NCT04888221	IL-6	52 weeks	Phase 3, randomized, parallel assignment, quadruple blind	PBO	GCA with cerebrovascular involvement	66	Percentage of patients in complete remission of GCA with absence of ischemic stroke recurrence at w24	Not yet recruiting
Tocilizumab [s.c. TCZ 162 mg/w 156 w or commercial availability of TCZ]	NCT03202368	IL-6	156 weeks	Phase 3, open-label extension of WA28119 (NCT01791153)	None	GCA flare or persistent disease activity	3	Percentage of subjects with adverse events at w160	Completed (not published)
Tocilizumab TCZ 8 mg/kg on day 3 and thereafter weekly s.c. TCZ injections (162 mg) over 52 w	GUSTO NCT03745586	IL-6	52 weeks	Open-label Phase 1 and Phase 2	GC	New-onset GCA	18	Analyze the effect of ultra-short GCs followed by TCZ monotherapy. Proportion of patients achieving remission within 31 days and without relapse until w24	Completed (final results not published yet)
Sarilumab	NCT03600805	IL-6	52 weeks	Randomized, parallel assignment, quadruple blind Phase 3	PBO	New-onset and refractory GCA	83	Proportion of patients with sustained remission at w52	Terminated (Protracted recruitment timeline exacerbated by COVID-19 pandemic)
JAK inhibitors									
Baricitinib [4 mg/d 52 w]	NCT03026504	JAK1+JAK2	52 weeks	Phase 2, open-label	None	Relapsing GCA	15	Percentage of subjects experiencing AE at w52	Completed (not published)
Upadacitinib [UPA dose A or dose B + 26 w GC taper versus PBO + 52 w GC taper]	SELECT-GCA NCT03725202	JAK1	52 weeks	Phase 3, randomized, parallel assignment, quadruple blind	PBO	New-onset and relapsing GCA	420	Percentage of patients achieving sustained remission at w52	Recruiting
IL-17 inhibitors									
Secukinumab [s.c. SEC 300 mg/4 w to w 48 + 26 w GC taper versus s.c. PBO to w 48 + 26 w GC taper]	TitAIN* NCT03765788	IL-17A	52 weeks	Randomized, parallel assignment, double-blind Phase 2	PBO	New-onset or relapsing GCA	52	Percentage of patients in sustained remission until w28	Completed (not published)
Secukinumab [s.c. SEC 300 mg/4 w 52 w + 26 w GC taper versus s.c. PBO + 52 w GC taper]	NCT04930094	IL-17A	52 weeks	Phase 3, randomized, parallel assignment, double-blind	PBO	New-onset or relapsing GCA	240	Number of participants with sustained remission at w52	Recruiting
Other drugs									
Anakinra [s.c. ANK 100 mg/d 16 w versus s.c. PBO/d 16 w]	GiAnT NCT02902731	IL-1	52 weeks	Phase 3, randomized, parallel assignment, double-blind	PBO	New-onset and relapsing GCA	70	Global relapse rate at w26	Recruiting
Abatacept [s.c. ABA 125 mg/w 12 m versus s.c. PBO/w 12 m]	ABAGART NCT04474847	CTLA-4 CD80/CD86	12 months	Phase 3, randomized, parallel assignment, double-blind	PBO	Newly diagnosed or relapsing GCA	78	Proportion of participants in remission of those randomized to ABA as compared to PBO at m12	Recruiting
Ustekinumab [s.c. UST 90 mg at w0, w4, w12 and w28 + GC taper versus GC taper]	ULTRA NCT03711448	IL-12/IL-23	52 weeks	Phase 2, randomized, parallel assignment, open-label	None	Relapsing GCA	38	Percentage of patients in remission, without a new relapse or deviation from the GC tapering protocol planned at w52	Recruiting
Guselkumab [i.v. GUS dose 1 at w0, w4 and w8 and s.c. GUS dose 2 q4w from w12 to w48 versus PBO, with 26 w GC taper in both arms]	THEIA NCT04633447	IL-23	52 weeks	Phase 2, randomized, parallel assignment, double-blind	PBO	New-onset or relapsing GCA	60	Percentage of participants achieving GC-free remission at w28	Recruiting
Mavrilimumab [s.c. MAV 150 mg q2w versus s.c. PBO, with 26 w GC taper in both arms] [105]	NCT03827018	GM-CSF	26 weeks	Phase 2, randomized, parallel assignment, quadruple blind	PBO	New-onset or relapsing GCA	70	Time to flare by w26	Completed (not published)
Bosentan [145 mg/d 14 d]	CECIBO NCT03841734	Endothelin receptors A and B	3 months	Phase 3, open-label	None	Sudden blindness due to GCA	8	Visual acuity calculated according to the Early Treatment Diabetic Retinopathy Study at m3	Unknown
Glucocorticoids [28 w GC taper versus 52 w GC taper]	CORTODOSE NCT04012905		52 weeks	Phase 3, randomized, parallel assignment, open-label	GC	New-onset GCA	150	Patients in complete remission throughout 52 w, without relapse	Not yet recruiting

Abbreviations: ABA: abatacept; AE: adverse event; AION: anterior ischemic optic neuropathy; ANK: anakinra; ASA: acetylsalicylic acid; CTLA-4: cytotoxic T-lymphocyte antigen 4; COVID-19: coronavirus disease 2019; d: day; g: gram; GC(s): glucocorticoids; GCA: giant cell arteritis; GM-CSF: granulocyte-macrophage colony-stimulating factor; GUS: guselkumab; IL: interleukin; IL-6R: interleukin 6 receptor; i.v.: intravenous; JAK: Janus kinase; kg: kilogram; m: month; MAV: mavrilimumab; mg: milligram; MP: methylprednisolone; n: number; PBO: placebo; qw: every week; q2w: every other week; s.c.: subcutaneous; SEC: secukinumab; TCZ: tocilizumab; UPA: upadacitinib; UST: ustekinumab; vs: versus; w: week. *Secukinumab (TitAIN): currently only the protocol has been published [97].

#### 3.3.4. Endothelin Receptor Antagonists

Endothelins (ET) are potent endogenous pressor agents, secreted by different tissues and cells of the body. The endothelin system includes a family of three peptides of 21 amino acids: endothelin-1 (ET1), endothelin-2 (ET2), and endothelin-3 (ET3). ET1 is synthesized by the vascular endothelium in response to a series of factors such as angiotensin II, insulin, hypoxia, and severe pressure decreases, although it is also synthesized by VSMCs. Endothelin-1 is a powerful vasoconstrictor, whose actions are mediated by two receptors, ETA and ETB [108]. ET1 also has a proliferative action on VSMCs, promotes fibroblast production, modulates extracellular matrix synthesis, causes VSMCs hypertrophy, affects vascular permeability, mediates inflammation, and stimulates the sympathetic nervous system [108].

Plasma ET1 concentrations were increased in patients with GCA and ischemic events, suggesting a possible correlation between ET1 and this complication. Additionally, increased levels of ET1 were also found in the temporal arteries of GCA patients. Treatment with i.v. GCs failed to decrease the ET1 concentration in tissue [109]. These results increase the potential relevance of ET1, especially in those patients with GCA and visual complications not responding to standardized conventional GC therapy [110].

Bosentan, an oral mixed endothelin receptor dual antagonist with affinity for ETA/ETB receptors, is currently under investigation in a phase 3, open label trial (NCT03841734), to assess its role in the management of blindness due to GCA (Table 3). Therefore, this constitutes a stimulating potential therapeutic opportunity for one of the most overwhelming complications of GCA [35].

## 4. Discussion

GCA is the most frequent medium-sized and large vessel vasculitis in the elderly population, especially of Caucasian origin. The clinical spectrum of the disease has been redefined in recent years. Although the classic or cranial form continues to be the predominant phenotype, we now know that there are other patterns that justify a more atypical presentation of the disease. In addition, the diagnosis of the disease has improved considerably in the last two decades. Although TAB continues to be the gold standard, imaging tests are fundamental pieces for an early diagnosis, as the ultrasound of the temporal and axillary arteries is one of the bases on which the diagnosis of the disease is currently carried out.

The main complication of GCA in the acute phase is visual loss, which can become permanent in 15–20% of patients, although its incidence fortunately has decreased in recent decades [111]. Conversely, the main long-term complications are vascular: aneurysms and stenosis of the large arteries, and aortic dissection, which may require surgery.

The current management of GCA remains suboptimal, with inadequate rates of maintenance of remission. For about 70 years, the treatment of GCA has been based almost exclusively on GCs. While effective at high doses, dose reduction frequently leads to disease relapses. GCs inhibit the Th17-dependent pathway, but not so much the Th-1-dependent pathway, on which ischemic complications and long-term arterial sequelae depend more. In fact, around 50% of patients relapse when the dose is reduced below 15–20 mg/day, especially the extracranial forms, so it is necessary to associate csDMARDs or biological therapy. Among the csDMARDs, MTX is the only one that seems to have a favorable relationship between efficacy and safety to reduce relapses and decrease the accumulated doses of GC in the long term, especially in patients who are dependent on GC or at a high risk of complications derived from GC use. In our opinion, MTX should be used at doses not lower than 15 mg/week, and possibly parenterally to achieve the desired objective. It is possible that other csDMARDs, such as cyclophosphamide or mycophenolate mofetil, may also be effective, but with a high risk of toxicity, particularly cyclophosphamide, which does not justify their current use.

TCZ, an IL-6 receptor inhibitor, has been demonstrated to increase the remission rate, reduce the number of recurrences, and achieve a lower cumulative dose of GC, making it currently the second line of therapy for GCA. In patients with symptoms of visual impairment, high-dose IV GC should be used. TCZ may be also considered in the first instance if there is already visual loss.

It is very likely that the arrival of TCZ biosimilars, which will considerably reduce the cost of this agent, will reorder the therapeutic strategy for GCA, making its use much more extensive and earlier in the course of the disease. However, despite this substantial advance in the management of GCA, many patients are unsuitable or unresponsive to TCZ and other IL-6 antagonists. In fact, nearly 50% of patients treated with TCZ in the GiACTA trial were not in sustained remission at week 52 [13]. Therefore, TCZ is not the “magic bullet” for all GCA patients, and we need other alternative therapeutic options.

Improvements in the understanding of the pathogenesis of the disease have opened up new roads and therapeutic opportunities in GCA. With the discovery of new agents, it seems that we are on the cusp of a new era in the management of GCA. There is currently an important portfolio of new therapeutic agents in different stages of development for patients who are refractory to GC, MTX and TCZ, or who develop side effects with these agents. Of these, the most promising paths at present are the inhibition of the IL12/IL23 axis, the inhibition of the IL17 pathway, the modulation of T cells with abatacept, the inhibition of GM-CSF and the inhibitors of the endothelin family, mainly ET1 inhibitors, especially when there are visual complications.

Similar to other autoimmune diseases, the future of GCA therapy should be based on a better understanding of the pathogenesis of disease, the heterogeneous clinical picture, the patient’s own profile, and their comorbidities. Only in this way will we move towards a personalized medicine, which is the basis for a more rational therapy of GCA and other immune-mediated diseases. 

## Figures and Tables

**Figure 1 jcm-11-01588-f001:**
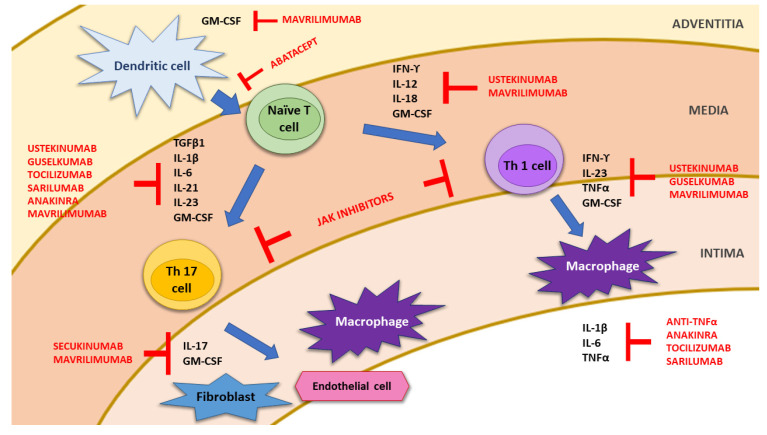
Schematic representation of the pathophysiology of giant cell arteritis. Abbreviations: GM-CSF: granulocyte-macrophage colony-stimulating factor; IL: interleukin; INF-Υ: interferon gamma; JAK: Janus kinase; T cell: T lymphocytes; Th: T helper lymphocytes subpopulation; TGF-β: transforming growth factor beta; TNF-α: tumor necrosis factor alpha.

**Table 1 jcm-11-01588-t001:** Tocilizumab in giant cell arteritis. Main observational studies including 15 or more patients and randomized clinical trials.

Author, Year [Ref.]	Type of Study	*n*	Sex F (%)	Population/Median Follow-Up (FU)	Therapeutic Protocol	Prior DMARD *n* (%)	Main Efficacy Outcomes	Serious Adverse Events
Observational Studies								
Loricera et al., 2015 [60]	Retrospective multicenter open-label study	22	17 (72.3)	Refractory GCA and/or with unacceptable GC-related AE. FU: 9 months	i.v. TCZ 8mg/kg/m (monotherapy *n* = 10, combined with MTX *n* = 12)	19 (86.4)	Clinical remission (17/22) or improvement (2/22). Significant reduction of ESR, CRP and GC dose. Prior visual loss not reverted by TCZ in 2 patients	TCZ discontinuation (*n* = 3) due to SAE (neutropenia, recurrent pneumonia and CMV). 1 Death (stroke and infective endocarditis)
Régent et al., 2016 [61]	Retrospective multicenter open-label study	34	27 (79.4)	GCA with unacceptable GC-related AEs (*n* = 31), severe disease (*n* = 2) or as a GC-sparing agent (*n* = 1). FU: 13 months	i.v. TCZ 8mg/kg/m (monotherapy *n* = 16, combined with MTX *n* = 18)	20 (58.8)	Clinical improvement (28/34). Significant reduction of CRP and GC dose. Prior visual loss not reverted by TCZ (*n* = 1). Planned stop of TCZ (*n* = 20), with 6 relapses after a mean of 3.5 m	TCZ discontinuation (*n* = 3) due to SAE (liver cytolysis, neutropenia and TBC pericarditis). 1 Death (septic shock)
Samson et al., 2018 [39]	Prospective multicenter open-label study	20	15 (75)	New-onset (95%) or recurrent GCA. FU: 52 weeks	4 Infusions of i.v. TCZ 8 mg/kg/m from w0 to w12	Prior DMARD within 6 m before inclusion were not allowed	Remission in 100% at w12 75% in remission with ≤0.1 mg/kg/d of GC at w26. 45% relapse-free survival at w52 Relapse (*n* = 10) more frequent if baseline aortitis (*p* = 0.048), CRP > 70 mg/L (*p* = 0.036) or Hb ≤ 10 g/dL (*p* = 0.015)	3 SAE: 1 sudden death for unknown reasons and 2 hospitalizations (atrial fibrillation and hip replacement)
Adler et al., 2019 [62]	Open-label extension of iv TCZ trial	17	13 (76.5)	Relapse-free patients from the TCZ arm. FU: 28.1 months	If relapse, TCZ or GC could be added at physician discretion	No GC or other DMARD were given after TCZ discontinua-tion	Lasting remission DMARD-free in > 50% of patients. 8 Patients relapsed: 6 within first 5 m and 2 at 13 m and 14 m, respectively. Relapsing patients were younger and had more signs of mural enhancement in MRA	NR
Nannini et al., 2019 [63]	Single-center prospective open-label study	15	NR	New-onset (*n* = 11) or recurrent GCA with thoracic LVV. FU: 18 months	i.v. TCZ 8mg/kg/m (*n* = 9) or s.c. TCZ 162 mg qw (*n* = 6) + GC taper of 2 m TCZ tapering from m6 to m10 TCZ interruption at m12	NR	100% Sustained remission during TCZ therapy. 66.7% Patients maintained remission after TCZ withdrawal. 5 Patients relapsed 2–4 m after TCZ interruption, responding to TCZ reintroduction at the last dose, allowing a second withdrawal of TCZ in 2 of them. Non-response correlated with some PET/CT parameters	No SAE
Calderón-Goercke et al., 2019 [64]	Multicenter retrospective series	134	101 (75.4)	Refractory GCA (100%)	i.v. TCZ 8mg/kg/m (*n* = 106) or s.c.TCZ 162 mg qw (*n* = 28) + GC	csDMARD 98 (73.1) bDMARD 3 (2.2)	Clinical remission was achieved in 55.5%, 70.4%, 69.2% and 90% of patients at 6, 12, 18, 24 months, respectively	SAE: 32 (21.1 per 100 patients-year). Serious infections: 16 (10.6 per 100 patients-year). TCZ withdrawals: 17 (12.7 per 100 patients-year)
Amsler et al., 2021 [65]	Observational monocentric study	186	116 (62)	MR angiography of aorta performed/ positive in: 170 (91%)/123 (72%). FU: NR	i.v. TCZ was added to GC in doses of 8 mg/kg body weight at 4-week intervals or at a dosage of 162 mg s.c. at weekly or bi-weekly intervals	NR	The occurrence of vision loss in a large GCA cohort treated with TCZ	Only visual AE described: Two patients (1.1%) developed vision loss, both at the initiation of TCZ treatment
Randomized clinical trials								
Villiger et al., 2016 [12]	Single center, phase 2, DB, PBO-controlled	30 (20 TCZ, 10 PBO) 2:1 ratio	TCZ 13 (65) PBO 8 (80)	New-onset (80% TCZ, 70% PBO) or relapsing GCA. FU: 52 weeks	i.v. TCZ 8mg/kg/m + GC vs. PBO + GC Same GC tapering schedule and concomitant drugs TCZ was used as mono-therapy	Prior bDMARD not allowed No mention about prior csDMARD	Complete remission at week 12: 85% TCZ vs. 40% PBO (RD 45%, 95% CI 11–79; *p* = 0.03). Complete remission at week 52: 85% TCZ vs. 20% PBO (RD 65%, 95% CI 36–94; *p* = 0.001). Significantly lower time to stop GC and cumulative GC dose in TCZ group vs. PBO group	35% TCZ vs. 50% PBO, (*p* = 0.46). TCZ: 1 headache and tinnitus, 3 GI (prepyloric ulcer perforation, viral hepatopathy and bleeding), 1 eye infection, 1 psychosis and 1 Stevens-Johnson syndrome. PBO: 3 CV events (syncope, coronary artery disease, lethal MI), 1 sigmoid perforation, 2 GC-induced myopathy and hyperglycemia, 2 back pain and 2 lumbar fractures and vertebroplasty
Stone et al., 2017 [13] Tuckwell et al., 2017 [66]	Multicenter, phase 3, DB, PBO-controlled (GiACTA trial)	251 (150 TCZ groups *, 101 PBO groups ^#^) 2:1:1:1 ratio	TCZ * 78 (78) 35 (70) PBO ^#^ 38 (76) 37 (73)	New-onset (47% and 52% TCZ groups *, 46% and 45% PBO groups ^#^) or relapsing GCA. FU: 52 weeks	s.c. TCZ 162 mg qw + GC 26 w (*n* = 100); s.c. TCZ 162 mg q2w + GC 26 w (*n* = 50); PBO + GC 26 w (*n* = 50); PBO + GC 52 w (*n* = 51). Concomitant MTX was the only DMARD allowed ^Ϯ^	Baseline MTX in 4 (3) newly diagnosed GCA and 23 (17) relapsing GCA	Sustained GC-free remission at week 52 significantly favored TCZ over PBO (*p* < 0.001): 56% TCZ qw, 53% TCZ q2w, 14% PBO + GC 26 w and 18% PBO + GC 52 w. Significantly lower risk of flare and cumulative GC dose in TCZ groups vs. PBO groups	15% TCZ qw, 14% TCZ q2w, 22% PBO + GC 26 w and 25% PBO + GC 52 w, NS. Infections were the most frequent SAE: 7%, 4%, 7% and 12%, respectively. 1 case of AION during a flare in TCZ q2w group

Abbreviations: AE(s): adverse event(s); AION: anterior ischemic optic neuropathy; bDMARD: biologic disease modifying antirheumatic drugs; CI: confidence interval; CMV: cytomegalovirus; CRP: C-reactive protein; csDMARD: conventional synthetic disease modifying antirheumatic drugs; CV: cardiovascular; DB: double-blind; DMARD: disease modifying antirheumatic drugs; ESR: erythrocyte sedimentation rate; F: female; FU: follow-up; GC: glucocorticoids; GCA: giant cell arteritis; GI: gastrointestinal; Hb: hemoglobin; i.v.: intravenous; kg: kilogram; LVV: large vessel vasculitis; m: month; mg: milligram; MI: myocardial infarction; MRA: magnetic resonance angiography; MTX: methotrexate; n: number; NS: not significant; PBO: placebo; PET/TC: 18-fluorodeoxyglucose (FDG)-positron emission tomography; qw: every week; q2w: every other week; RD: risk difference; SAE: serious adverse events; s.c.: subcutaneous; TBC: tuberculosis; TCZ: tocilizumab; vs: versus; w: week. * TCZ groups: s.c. TCZ administered every week or every other week; ^#^ PBO groups: with a GC tapering protocol of 26 weeks or 52 weeks. ^Ϯ^ MTX allowed: stable doses of concomitant MTX were allowed by protocol if started more than 6 weeks prior to the study enrollment, and maintained stable throughout the screening and 52-week double-blind treatment periods. The rest of DMARD was excluded by protocol.

## Data Availability

Not applicable.

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
