# Peer review of "Advances in the Treatment of Giant Cell Arteritis"

_jcm, 2022, doi:10.3390/jcm11061588_

Round 1

Reviewer 1 Report

I read with interest this review about therapeutic options for GCA and I found it complete and very clear. The main literature was examined and correctly reported.

Author Response

Reviewer#1:

I read with interest this review about therapeutic options for GCA and I found it complete and very clear. The main literature was examined and correctly reported.

Response (R): We thank reviewer#1 for his/her positive commentaries.

Reviewer 2 Report

In general, this paper presents a very good overview of the current forms of therapy for giant cell arteritis. It systematically addresses the different targets of the individual pharmaceutical drugs and explains them in the context of the disease. Corresponding studies with a focus on the currently applied therapy as well as the corresponding therapy protocols are also shown. The manuscript is well structured and a very well-researched work. 

Author Response

Reviewer#2:

In general, this paper presents a very good overview of the current forms of therapy for giant cell arteritis. It systematically addresses the different targets of the individual pharmaceutical drugs and explains them in the context of the disease. Corresponding studies with a focus on the currently applied therapy as well as the corresponding therapy protocols are also shown. The manuscript is well structured and a very well-researched work.

Response (R): We thank reviewer#2 for his/her favorable opinion and comments.

Reviewer 3 Report

The authors reviewed the therapeutic advances in giant cell arteritis. The manuscript is up to date, clear and well-written.

I would only have very few comments:

- Although the manuscript is well-written, several typos remained and requires a careful English editing (mayor, tozilizumab, "all the patients MRA mural signal at the end of follow-up despite being in remission", biospy, ...)

- The sentence “IL-12 and IL-23 are two cytokines central to the inflammatory and proliferative path-517 ways of GCA (Figure 1) [84,85].”is repeated (line 217)

- ABA not defined in the Table 2 legend, and the abbreviation for abatacept is sometimes ABA, sometimes ABT

- “Additionally, adalimumab was associated with an increased risk of serious side effects” does not seem to be supported by the data of the study (14.7 vs 47.2%, respectively) (line 508).

- “Anakinra (ANK), a recombinant form of human IL-1 receptor antagonist, proved to be effective in the management of GCA in three patients with refractory GCA”. The efficacy of this treatment has now been reported in 6 patients, including 4 new patients (doi: 10.1093/rheumatology/keab280)

Author Response

Reviewer#3:

The authors reviewed the therapeutic advances in giant cell arteritis. The manuscript is up to date, clear and well-written.

I would only have very few comments:

- Although the manuscript is well-written, several typos remained and requires a careful English editing (mayor, tozilizumab, "all the patients MRA mural signal at the end of follow-up despite being in remission", biopsy ...)

Response (R): The reviewer is absolutely right. Therefore, typographical mistakes have been corrected and a careful English editing has been made by an official translator of our institution.

- The sentence “IL-12 and IL-23 are two cytokines central to the inflammatory and proliferative path-517 ways of GCA (Figure 1) [84,85].”is repeated (line 217)

R: The reviewer is completely right, there were two identical consecutive sentences. Therefore, I have deleted one of them.

- ABA not defined in the Table 2 legend, and the abbreviation for abatacept is sometimes ABA, sometimes ABT

R: The reviewer is right again. Therefore, we have added ABA to the end of Table 2 as a footnote, and replaced all ABTs with ABA between lines 580 and 595 of the text.

- “Additionally, adalimumab was associated with an increased risk of serious side effects” does not seem to be supported by the data of the study (14.7 vs 47.2%, respectively) (line 508).

R: We thank the reviewer for his/her truthful comment. In fact, the percentages of adverse effects were completely reversed. Accordingly, we have corrected and changed them.

- “Anakinra (ANK), a recombinant form of human IL-1 receptor antagonist, proved to be effective in the management of GCA in three patients with refractory GCA”. The efficacy of this treatment has now been reported in 6 patients, including 4 new patients (doi: 10.1093/rheumatology/keab280).

R: The reviewer is right once again, which is why we have now included this ANK study as a GCA treatment in both the Discussion (lines 610-617) and the references. The new reference included has been numbered as reference 96, the original reference 96 being removed.
